# The carbon perception gap in actual and ideal carbon footprints across wealth groups

Johanna Köchling [1,2,6] ✉, Julia E. Koller [1,6], Jana Straßheim [1,2], Yannic Rehm[3], Lucas Chancel [3,4], Claudia Diehl [5], Harald T. Schupp [1,2] & Britta Renner [1,2]

Carbon inequality is gaining attention in public discussions surrounding equitable climate policies. It commonly refers to the unequal distribution of individual greenhouse gas emissions, with wealthier individuals contributing disproportionately higher emissions. Little is known about how people perceive the actual carbon footprint distribution across wealth groups and what they would desire as an ideal distribution. Survey data from Germany show awareness of carbon inequality, with respondents recognizing that wealthier individuals emit disproportionately more. Yet, with surprising consensus, all groups, including the wealthy, desired an inverse ideal distribution, with the wealthy having disproportionately smaller carbon footprints. Nonetheless, most perceived their own carbon footprint as far better compared to others in society and within their wealth group. Here, we show a carbon perception gap, particularly among the wealthiest: Collectively, people acknowledge the presence of carbon inequality and desire a more equitable distribution, yet often perceive themselves as already contributing more than others.

In the wake of the climate crisis, humanity faces an increasingly urgent need to cut greenhouse gas (GHG) emissions. Yet, not all humans contribute equally to the climate crisis. Estimates suggest that the wealthiest 10% of the global population contribute approximately 45–49% of global GHG emissions, while the bottom 50% contribute only 7–13%[1–4] (for income based estimates, see also Starr et al.[5]). In Germany, which is characterized by one of the highest levels of wealth inequality in the Eurozone[6], the wealthiest 20% of the population are responsible for 25–52% of the country's GHG emissions depending on the accounting framework, whereas the poorest 20% emit only 8–16%[7] (see "Methods" section for detailed methodology). This disparity is particularly significant as Germany remains the EU's top GHG emitter, producing 750 million metric tons of GHG emissions in 2022, despite its ambitious climate targets to decrease GHG emissions by at least 65% until 2030[8,9]. This combination of wealth disparity, high GHG emissions, and ambitious goals poses substantial challenges for equitable climate action, particularly given Germany's influential role as a key

economic and political player in the EU. However, achieving more equitable contributions remains a challenge faced by many countries and regions.

Carbon inequality is at the center of debates and disagreements about equitable climate policies because it raises the question of who should contribute to climate change mitigation and to what extent. How much GHG emissions should groups in society with greater financial resources ("wealthier groups") be entitled to emit compared to those with fewer resources ("poorer groups")? Determining the ideal societal distribution of carbon footprints is complex and a source of friction between stakeholders. Although people generally view equality as an important principle in the abstract[10–12], research on the ideal distributions of wealth[13,14] and health[15] has shown that people are willing to accept inequalities to a certain extent. Thus, people may not necessarily want everyone to contribute equally, as some levels of inequality in carbon footprints might even be considered desirable. For example, people may find

[1]Department of Psychology, University of Konstanz, Konstanz, Germany. [2]Centre for the Advanced Study of Collective Behaviour, University of Konstanz, Konstanz, Germany. [3]Paris School of Economics, Paris, France. [4]Center for Research on Social Inequalities, Sciences Po, Paris, France. [5]Department of Sociology, University of Konstanz, Konstanz, Germany. [6]These authors contributed equally: Johanna Köchling, Julia E. Koller. ✉ e-mail: johanna.koechling@uni-konstanz.de

some form of carbon inequality acceptable because they believe that wealthier groups contribute more to economic growth and are therefore entitled to consume and emit more. Conversely, it could be argued that wealthier groups have more resources and opportunities to reduce their GHG emissions and should therefore emit even less than poorer groups[16]. However, the desired distribution of carbon footprints across different wealth groups in society remains unknown.

Beyond the ideal distribution of carbon footprints across wealthier and poorer groups in society, the perception of current inequalities is considered an essential driver of support for structural and redistributive policies[10,17,18], potentially even more so than the actual level of inequality[19–21]. For example, informing individuals about the objective distribution of household GHG emissions across different income groups, as well as their own position within the income distribution, increased support for carbon taxation, particularly among less affluent groups[22]. There is emerging evidence of an underestimation of carbon inequality[23,24]. When asked who causes more environmental harm (including pollution and global warming), less than 50% of Brazilian respondents identified the rich or higher income groups as causing more environmental harm[23]. Furthermore, comparing carbon footprint estimates for three income groups (bottom 50%, top 10%, top 1%) from online samples in Denmark, India, Nigeria, and the USA revealed that participants attributed higher emissions to the top income groups but generally underestimated the gap between the top 1% and bottom 50%[24]. However, perceptions of the carbon footprint distribution across all wealth groups, including the poor, remain unclear. Additionally, discrepancies between ideal and actual carbon footprint distributions provide insights into whether individuals are satisfied with the status quo or see a need for change.

Finally, to understand perceptions of carbon inequality, it is crucial to consider how people perceive their own carbon footprint. Although wealthier individuals typically have larger carbon footprints, they spend a smaller proportion of their income on GHG-intensive goods compared to less affluent individuals[25]. Additionally, individuals may view their own GHG emissions more favorably than those of others, which could be particularly pronounced among the wealthy who tend to have higher GHG emissions. However, it remains to be studied whether individuals account for wealth when evaluating their own and others' carbon footprints.

Our study's objectives are therefore threefold: (1) we investigate perceptions of ideal and actual carbon footprint distributions across five wealth groups (i.e., quintiles) in the German population. (2) Additionally, we assess whether these perceptions are grounded in reality by comparing them to three different objective carbon footprint estimates for the same five wealth groups. These estimates are based on the consumption-based approach (i.e., all GHG emissions attributed to final consumers), the mixed and ownership-based approach (i.e., GHG emissions split between final consumers and production owners), as described by Chancel and Rehm[7]. (3) Finally, we examine how individuals in each wealth group perceive their own personal carbon footprint.

Here we show a carbon perception gap, particularly among the wealthiest. While people acknowledge carbon inequality and express a desire for a more equitable distribution, they believe they are already contributing more than others. Thus, people seem to acknowledge the existence of carbon inequality and share the goal of greater equity, but perceive themselves to be ahead of others in contributing to these goals. Therefore, our study has implications for determining which carbon footprint distributions are considered as deemed just and how closely these perceptions align with perceptions of actual distributions, which is important for identifying possible solutions and generating support for structural and redistributive policies.

## Results

### Perceptions of ideal carbon footprint distributions

Most participants considered a good carbon footprint (i.e., low GHG emissions) to be important (83.8%). This is also reflected in their ideal distribution of carbon footprints across the five national wealth quintiles as illustrated in Fig. 1a.

The vast majority of the participants viewed a very good or good carbon footprint (i.e., very low or low GHG emissions) as desirable (see Fig. 1a). At least 78.5% of the participants preferred a very good or good carbon footprint for each national wealth quintile. However, they did not favor an entirely equal distribution across the five national wealth quintiles ($b = 0.10$, 95% CI [0.08; 0.11], $t(1375.58) = 15.13$, $p < 0.001$; see Table 1, Supplementary Table 1), with national wealth quintiles explaining 64.7% of the observed variance. The most ambitious ideal carbon footprint (i.e., lowest GHG emissions) was construed for the wealthiest quintile, with 91.8% of participants preferring the top 20% to have a very good or good carbon footprint. By contrast, only 78.5% expected the poorest quintile to ideally have a very good or good carbon footprint.

As shown in Fig. 2, participants from all wealth quintiles, including those at the top, desired better carbon footprints (i.e., lower GHG emissions) by the rich compared to the poor. Notably, there was a strong consensus for this, with 93.8% positive random slopes across participants ($range_{Positive\ RS} = 0.97 \geq b \geq 0.01$). However, the extent of this preference varied: wealthier participants envisaged a more ambitious ideal distribution, favoring better carbon footprints across wealth quintiles, including their own (personal wealth quintile: $b = 0.05$, 95% CI [0.01; 0.09], $t(1377.44) = 2.73$, $p = 0.006$; national x personal wealth quintile: $b = -0.01$, 95% CI [−0.03; 0.00], $t(1375.56) = -2.09$, $p = 0.037$; see Supplementary Table 1). Comparing ratings from participants in the top and bottom 20% illustrates this effect (Fig. 2). For example, 82.4% participants belonging to the top 20% believed that the top 20% of the German population should have a very good carbon footprint, compared to 71.1% of the bottom personal wealth quintile. Similarly, 67.6% of the top and 55.6% of the bottom personal wealth quintile believed that the poorest 20% of the German population should have a very good carbon footprint.

### Perceptions of actual carbon footprint distributions

In contrast to the ambitious ideal distributions, participants' perceptions of the actual carbon footprint distribution in Germany were far more negative. Only 27.3% to 9.9% of the participants believed that any wealth quintile currently achieves a very good or good carbon footprint (Fig. 1b). The wealthiest 20% were seen as having the worst carbon footprint, reflecting an awareness of carbon inequality in Germany ($b = -0.26$, 95% CI [−0.28; −0.24], $t(1378.28) = -20.81$, $p < 0.001$; Pseudo-$R^2 = 0.61$; Table 1 and Supplementary Table 4). Specifically, 78.4% estimated the richest quintile's carbon footprint as very bad or bad (i.e., very high or high GHG emissions), compared to 43.9% for the poorest wealth group.

There was a strong consensus that the wealthy have a worse carbon footprint than the poor, with 79.7% negative random slopes across participants ($range_{Negative\ RS} = -1.03 \leq b \leq -0.03$). The majority of participants across all five personal wealth groups, including the richest, perceived a significant carbon inequality (Supplementary Fig. 2). This perception was not modulated by personal wealth ($b < 0.01$, 95% CI [−0.04; 0.04], $t(1378.39) = 0.09$, $p = 0.928$) or the interaction between personal wealth quintile and national wealth quintile ($b = 0.02$, 95% CI [0.00; 0.05], $t(1379.30) = 1.60$, $p = 0.110$; Table 1, Supplementary Table 4).

### Perceptions of personal and others' carbon footprint

Respondents evaluated their own carbon footprint clearly more favorably compared to their perceptions of the actual carbon footprint distribution across the population (Fig. 1c). Across the five personal

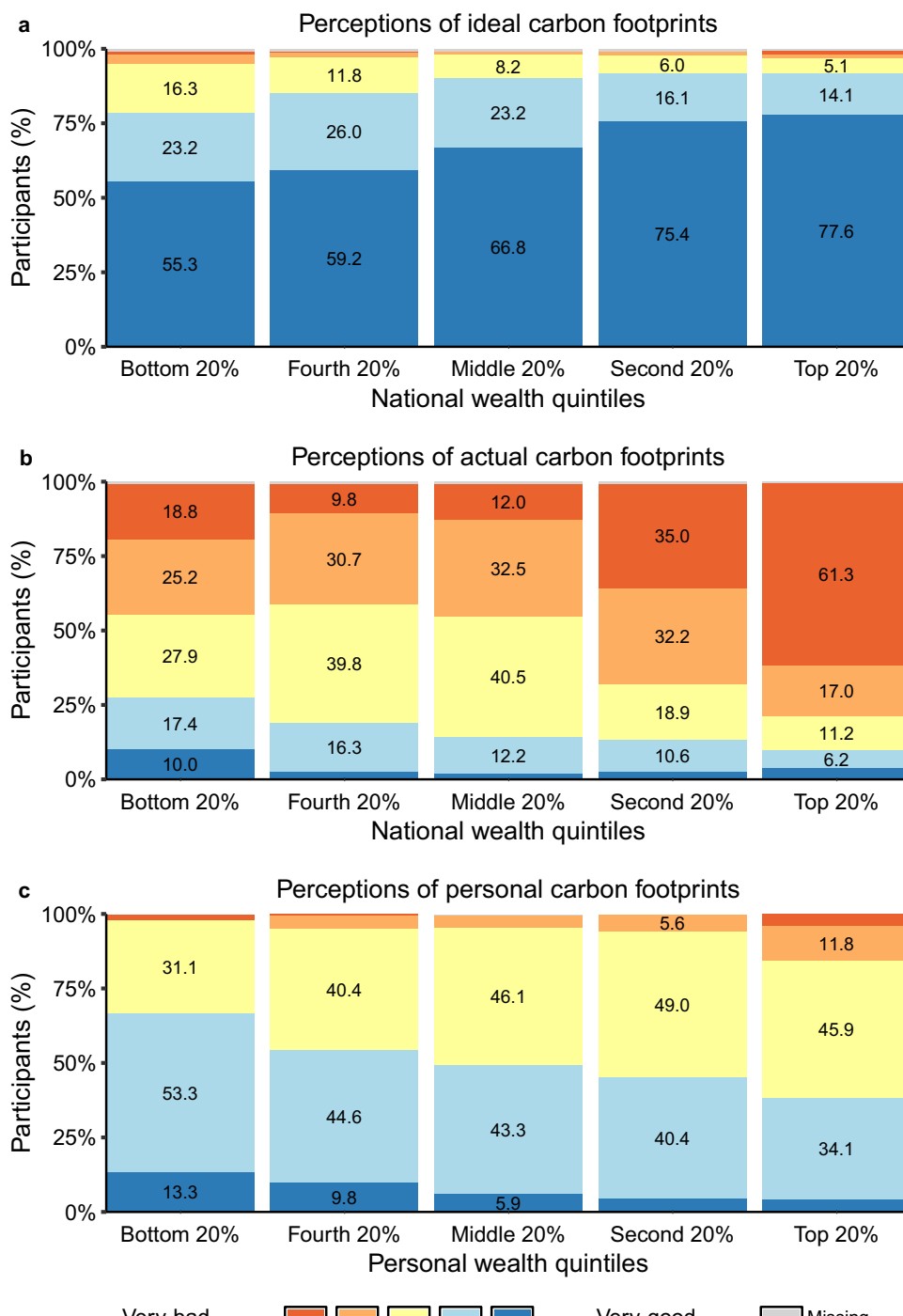

**Fig. 1 | Perceptions of ideal, actual and personal carbon footprints by wealth quintiles. a** Percentage (%) of participants' ratings of the ideal carbon footprint for each of the five national wealth quintiles (irrespective of their personal wealth quintile; top 20%: $n = 1378$ (8), second 20% and middle 20%: $n = 1376$ (10), fourth 20%: $n = 1374$ (12), bottom 20%: $n = 1375$ (11); ratings with missing values in parenthesis). **b** Percentage of participants' ratings of the actual carbon footprint of each national wealth quintile (irrespective of their personal wealth quintile; top 20%: $n = 1378$ (8), second–bottom 20%: $n = 1375$ (11); ratings with missing values in parentheses). **c** Percentage of participants within each personal wealth quintile rating their personal carbon footprint ($n = 1382$; 4 missing). Ratings were assessed on a scale from very bad (red) to very good (dark blue). Values below 5%, including missing values, are not labeled for presentation purposes.

wealth quintiles, 66.7% to 38.2% believed their carbon footprint was good or very good, while only 15.9% to 2.2% rated theirs as bad or very bad. Notably, wealthier participants rated both their personal carbon footprint and that of their own wealth group less positively (personal wealth quintile: $b = -0.11$, 95% CI [−0.15; −0.08], $t(1380) = -5.67$, $p < 0.001$; $R^2 = 0.02$; $r = -0.15$, $p < 0.001$; Supplementary Table 7 and Supplementary Fig. 3).

Furthermore, as shown in Fig. 3 (see also Supplementary Fig. 1 and Supplementary Table 8), participants consistently rated their own carbon footprint as far better than that of others in their wealth quintile (in-group comparison) and more closely aligned with their ideal. This pattern of carbon perception gap is particularly evident when comparing the three perceptions—ideal, actual and personal—within the top 20% wealth quintile. While 58.8% rated the carbon

**Table 1 | Results from multilevel models predicting participants' perceptions of ideal and actual carbon footprints**

| | Ideal | | Actual | |
|---|---|---|---|---|
| | Model 1 | Model 2 | Model 1 | Model 2 |
| Intercept | 4.52, $p < 0.001$ | 4.51, $p < 0.001$ | 2.38, $p < 0.001$ | 2.38, $p < 0.001$ |
| National wealth quintile | 0.10, $p < 0.001$ | 0.10, $p < 0.001$ | −0.26, $p < 0.001$ | −0.27, $p < 0.001$ |
| Personal wealth quintile | | 0.05, $p = 0.006$ | | 0.00, $p = 0.928$ |
| National × personal wealth quintile | | −0.01, $p = 0.037$ | | 0.02, $p = 0.110$ |
| Pseudo-$R^2$ | 0.65 | 0.65 | 0.61 | 0.61 |

Results from multilevel analyses (two-sided) are displayed for the preferred random slopes models. The predictors "national wealth quintile" and "personal wealth quintile" were recoded so that the intercept represents the predicted ideal/actual carbon footprint for the middle national/personal wealth quintile. All coefficients are unstandardized.

footprint of others in their quintile as very bad (Fig. 3b), they believed their own footprint was better (Fig. 3c) and closer to their ideal (Fig. 3a).

**Comparison between perceived and objective carbon footprints**
In a final step, we investigated whether participants' perceptions of ideal and actual carbon footprint distributions in society were rooted in reality. We compared the relative carbon footprint contributions attributed to the five national wealth quintiles with objective estimates (see Fig. 4).

For these objective estimations, we used three different approaches (see "Methods" section and Chancel and Rehm[7]). The consumption approach, the most commonly used approach for determining individual carbon footprints, allocates all GHG emissions to final consumers. Additionally, we applied the mixed and ownership approaches, which attribute GHG emissions to varying degrees to consumers and owners of productive assets. All three approaches yield carbon footprint estimates that increase with wealth and are consistent in their rank order of allocating GHG emissions to wealth quintiles.

Figure 4 illustrates the differences between perceived ideal and objective carbon footprint shares. Wealthier quintiles were desired to have smaller carbon footprint shares, compared to objective estimates. For the top wealth quintile, objective estimates were 1.7 to 3.5 times larger than the ideal. In contrast, for the bottom wealth quintile, objectively estimated shares were 0.6 to 0.3 times smaller than the participants' ideal. This pattern was consistent across personal wealth quintiles (Supplementary Fig. 4).

Perceived distributions of actual carbon footprints matched objective consumption-based estimates, but differed profoundly from objective estimates using the mixed and ownership approaches. The top quintile's carbon footprint share was underestimated by a factor of 0.7 to 0.5, while the bottom wealth quintile's share was overestimated by a factor of 1.3 to 2.3. This pattern also held across personal wealth quintiles (Supplementary Fig. 4).

## Discussion
The present study examined perceptions of ideal and actual carbon footprint distributions across five national wealth quintiles. Overall, there was a strong preference for very good carbon footprints (i.e., very low GHG emissions), with the desire that the rich, in particular, should have a very good carbon footprint. However, perceptions of actual carbon footprints in society showed the opposite: not only were they generally seen as markedly worse than the ideal, but also as particularly bad (i.e., very high GHG emissions) among the rich. In contrast, participants in all wealth quintiles rated their personal carbon footprint far better than their perception of the carbon footprint of others would suggest. This was particularly striking among participants in the top 20% wealth group.

Remarkably, the perceived distribution of actual carbon footprints appears to be highly similar to objective consumption-based carbon footprints estimated for the five national wealth quintiles. This

suggests that participants had a fairly accurate understanding of how GHG emissions are distributed among wealth groups in Germany, recognizing the status quo as characterized by high GHG emissions and carbon inequality. By contrast, carbon inequality is clearly underestimated compared to objective estimates that take production ownership into account (i.e., mixed and ownership estimates)[23,24]. Using objective estimates including public and private investments, a study by Nielsen et al.[24] also observed such an underestimation in online samples from Denmark, India, Nigeria, and the United States. However, whether perceptions of carbon inequality are deemed accurate or not critically depends on the objective standard of comparison used to probe for accuracy. Importantly, the different objective estimates used for comparison in this study can shed light on individuals' reasoning underlying their perceptions of the actual carbon footprint distribution. That is, individuals currently seem to attribute GHG emissions in a way that aligns with the consumption-based approach, which has dominated public discourse over the last decades, while potentially overlooking the less visible responsibility of production owners for GHG emissions (see Chancel and Rehm[7] for a discussion of consumers' and owners' responsibilities). Future research should explore how individuals intuitively assess the distribution of carbon footprints and assign responsibility for various emission-related factors, particularly those with differing levels of visibility in society. Moreover, it may be insightful to observe changes in perceptions when individuals are informed about the objective level of carbon inequality as estimated by different approaches.

Both the perceived and objective carbon inequality stand in stark contrast to the ideal carbon footprint distribution, suggesting a desire to change the perceived status quo[17]. There was a broad consensus among participants that GHG emissions are too high and must be significantly reduced. This insight is highly significant, as it shows that societal goals may be less controversial than often portrayed in public debates[26]. Moreover, the data clearly indicate that equal contributions are not desired, contradicting potential affirmative responding. Instead, participants across all wealth quintiles placed greater responsibility for GHG emissions on wealthier groups and expected them to be at the forefront of emission reductions and climate protection (see the polluter-pays principle[27,28]). The present results are supported by previous research showing that, for example, structural climate policies are especially desired from wealthier countries[29] and that carbon taxes favoring poor and middle-income households are preferred[26]. Together, these findings suggest an awareness of the decisive role wealth plays in creating opportunities for reducing GHG emissions. For instance, Kukowski and Garnett[16] recently argued that choosing low-carbon options requires time, financial resources, and access to supportive infrastructure, all of which poorer individuals often lack to a greater extent. This perspective appears to inform how individuals consider the roles of different actors and stakeholders (e.g., countries, individuals across wealth quintiles) in contributing to GHG emissions and climate protection[30].

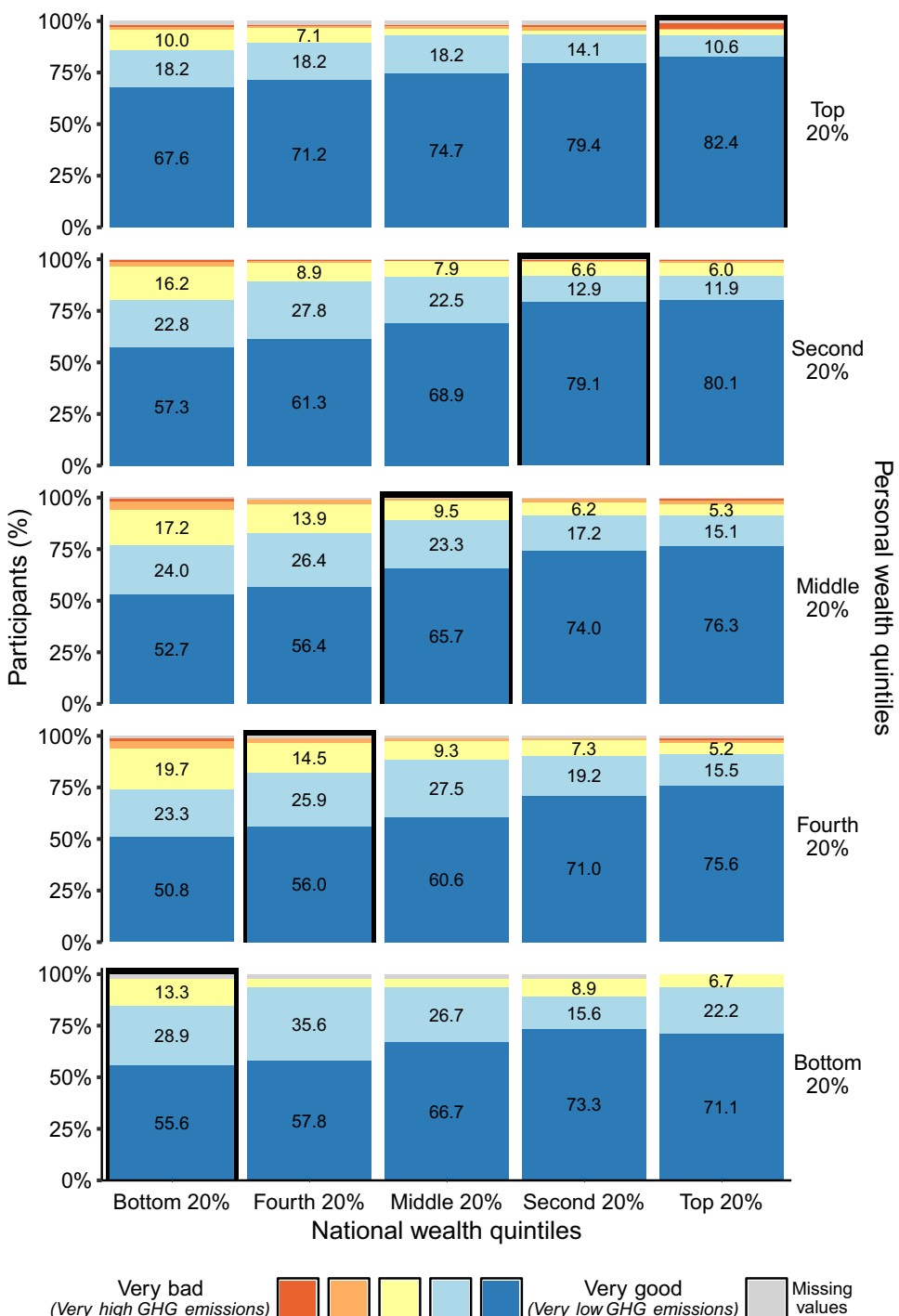

**Fig. 2 | Perceptions of ideal carbon footprint distributions across national wealth quintiles by personal wealth quintiles.** Percentage (%) of participants who rated the ideal carbon footprint for each of the five national wealth quintiles, from very bad (red) to very good (dark blue), grouped by their personal wealth quintile. Black frames highlight ratings of others' carbon footprints within the participant's own wealth quintile (in-group ratings). For example, 82.4% of individuals in the top 20% personal wealth group believe that their group should have a very good carbon footprint (i.e., very low GHG emissions; top right corner). Values below 5%, including missing values, are not labeled for presentation reasons. The number of participants per personal wealth quintile, with missing values in parenthesis, is as follows: top 20%: 170 (2–3), second 20%: 302 (1), middle 20%: 674 (3–5), fourth 20%: 193 (2), bottom 20%: 45 (0–1).

Compared to perceptions of ideal and actual carbon footprint distributions in society, participants from all wealth quintiles viewed their personal carbon footprint as being already closer to the ideal than the actual carbon footprints of others in society. This perception held even among the richest participants, although, in line with the presence of carbon inequality, they rated their personal carbon footprint as relatively worse than participants within other wealth quintiles, suggesting a degree of relative accuracy[31]. While selective sampling of participants with above-average carbon footprints cannot be entirely ruled out, given that we did not estimate participants' personal carbon footprints objectively, this is unlikely to fully explain the generally positive perceptions of personal carbon footprints at the group level[32]. Favorable perceptions of personal carbon footprints relative to that of others' suggest that these perceptions may not be fully grounded in

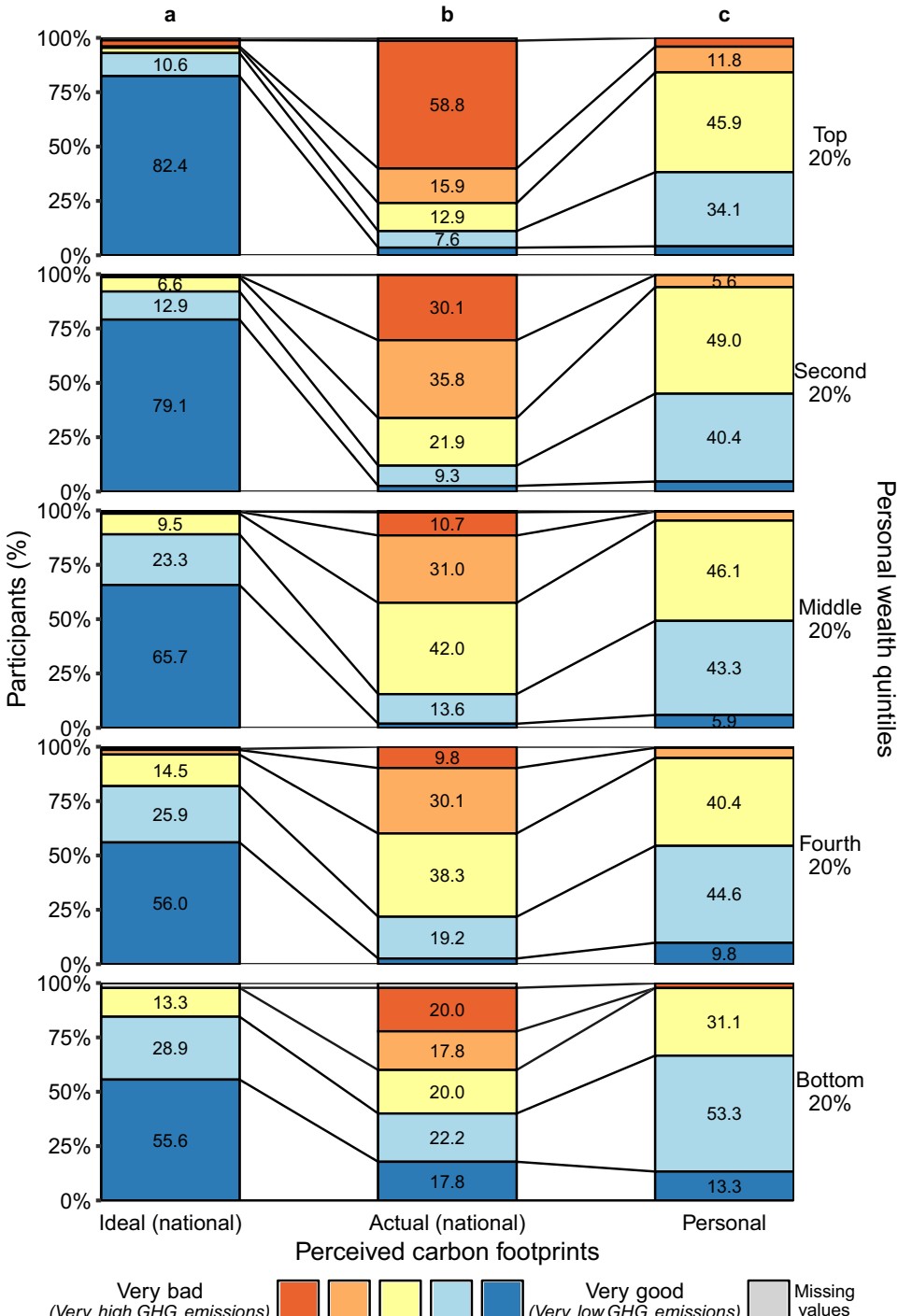

**Fig. 3 | Perceptions of ideal, actual and personal carbon footprints within the five personal wealth quintiles. a** Participants' ratings of ideal carbon footprints for others within their own wealth quintile. **b** Participants' ratings of actual carbon footprints for others within their wealth quintile. **c** Participants' self-rating of their personal carbon footprint (perceived personal carbon footprint). Displayed are the percentages (%) of participants who rated ideal, actual and personal carbon footprints on a scale from very bad (red) to very good (dark blue). The number of participants per personal wealth quintile, with missing values in parenthesis, is as follows: top 20%: 170 (0–2), second 20%: 302 (0–1), middle 20%: 674 (2–5), fourth 20%: 193 (0–2), bottom 20%: 45 (0–1). Values below 5%, including missing values, are not labeled for presentation reasons.

reality[33], potentially reflecting an optimistic bias[34,35] and the better-than-average effect[36,37].

These biases can stem from both cognitive and motivational mechanisms. A cognitive explanation is that forming accurate judgements about personal carbon footprints is challenging, especially given that GHG emissions from individual behaviors can vary significantly[38,39]. People may not always be aware of the most significant contributors to their carbon footprint (e.g., following a vegan diet)[40] and might overlook certain GHG emissions associated with less visible choices they make (e.g., GHG emissions linked to investment choices). Given that participants selectively viewed their own carbon footprint rather positively—regardless of wealth—we assume that they underestimated their own carbon footprint due to self-serving motivated reasoning about their behaviors and control

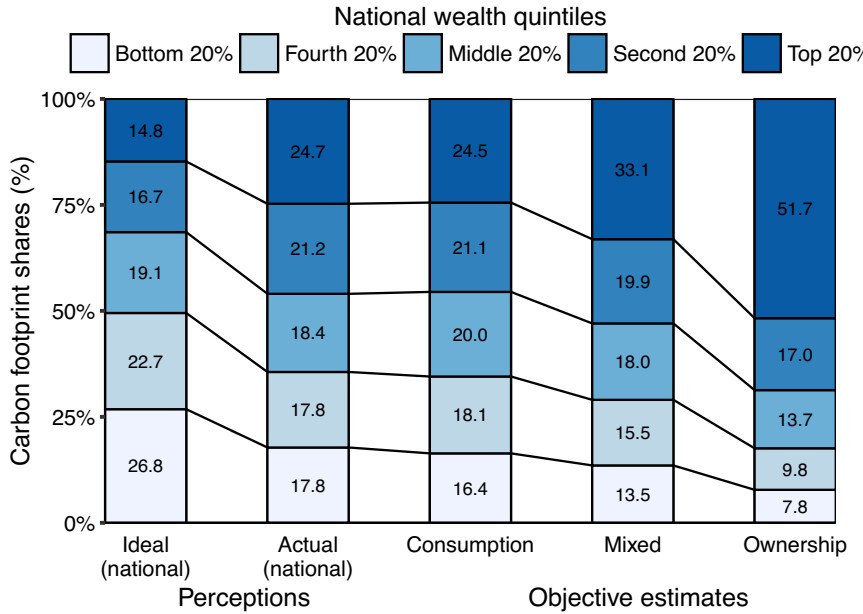

**Fig. 4 | Perceived and objective carbon footprint distributions.** Higher shares indicate a worse carbon footprint (i.e., higher GHG emissions) for the respective national wealth quintile (bottom 20% light blue–top 20% dark blue). Perceptions of ideal (*n* = 1174) and actual (*n* = 1374) carbon footprints of each of the five wealth quintiles were reverse-coded, divided by the total sum of these ratings, and converted into percentages (%). For the ideal and actual shares, 12 participants had to be excluded due to missing values on one or more items.

capabilities. This reasoning allows individuals to justify their optimistic beliefs about their own contribution while holding comparably negative beliefs about others[32,41,42]. For example, although we did not assess the specific behaviors participants considered when estimating their carbon footprint, we assume they selectively focused on their own climate-friendly behaviors. This tendency of selective information activation and retrieval from memory is well documented in psychology[43]. Numerous studies have shown that people aim to maintain positive self-perceptions through motivated reasoning and biased information processing[41,44,45]—a finding observed across cultures, albeit to varying extents[46,47].

Consistent with the motivated reasoning account[48,49], individuals do not need to bias all estimates to protect their self-esteem or positive self-view. Rather, they may selectively adjust perceptions that support their self-concept or goals while still acknowledging some level of reality. For example, participants did not consistently view their own group's carbon footprint more favorably than other groups; those in the first and second quintile showed a slight tendency toward in-group bias, rating their own group's carbon footprint somewhat more positively than others'. By balancing self-serving biases with an acceptance of certain realities, individuals may psychologically benefit from recognizing that GHG emissions are too high and that ambitious targets are needed—particularly for those with higher GHG emissions—while still believing they are personally more advanced than others in reducing their carbon footprint.

Regardless of the specific causes, overly optimistic perceptions of personal carbon footprints may affect behavior and the perception of public policies. That is, individuals with comparatively positive perceptions may be less inclined to reduce their carbon footprint[37] and may view climate policies that affect themselves as less fair[27,28]. This highlights the potential benefit of improving the accuracy of individuals' perceived carbon footprints, for instance, by providing personalized feedback[50] on their carbon footprints or by using descriptive and injunctive normative messaging to address the accuracy of their perceptions of their own behavior relative to others[37]. However, evidence regarding the effectiveness of these approaches remains mixed, and further research is needed[51–55]. Integrating more advanced

methods or questionnaires—such as personal tracking of energy use, transportation, diet, or asset ownership—to estimate individual-level objective carbon footprints would be a valuable future step in improving the accuracy of perceptions and understanding individual differences in emission patterns. However, it appears unlikely that it would alter the core conclusions: People consistently believe that carbon footprints are too large, should be reduced, but that they are already contributing more than others.

While the present study deepens our understanding of carbon inequality perceptions, some limitations should be noted. First, we assessed personal wealth quintiles subjectively, which could be biased[10,56]. In fact, when comparing individuals' self-categorizations to wealth quintiles and numeric wealth groups, we observe a tendency toward the middle wealth quintile (Supplementary Table 9; see also center bias[57,58]). Yet, how individuals perceive their financial position in society is relevant for studying inequality and may even be more relevant in terms of their policy preferences than their actual position[56,59]. Second, it has to be noted that the comparison between perceived carbon footprints and objective estimates of GHG emission shares across the five wealth quintiles is based on different metrics. To more conclusively relate perceived carbon footprints to objective carbon footprint estimates, measures using comparable metrics (e.g., asking respondents about the GHG emission shares of societal wealth groups) would be desirable. Third, while using the same scale across actual, ideal, and personal ratings reduces measurement errors and ensures comparability, potential biases stemming from this survey design cannot be ruled out entirely. However, structural bias seems unlikely as an established questionnaire method was used[13–15] and response patterns did not indicate misinterpretation or varying interpretations of question wording. Given the strong consistency in the pattern of results, it is likely that the findings reliably reflect respondents' perceptions. In addition, participants' perceptions did not appear generally biased. Perceptions of actual carbon footprint distributions closely aligned with objective estimates derived from the consumption-based approach. This suggests that participants had a fairly accurate understanding of how carbon footprints are distributed among wealth groups in Germany, despite the absence of additional

information on the impact of specific behaviors. This alignment further supports the validity of the survey design and the observed response pattern. Fourth, we assessed carbon footprint distributions using an indirect method, with respondents rating actual and ideal carbon footprints separately on identical scales. While a direct method (i.e., asking participants to specify whether the ideal carbon footprint should be smaller or larger than the actual) is more efficient and reduces participant workload, it may introduce social desirability bias, as participants could adjust responses to align with perceived expectations[34,47]. The indirect method minimizes this risk and allows for a more precise calculation of perception gaps by independently assessing both constructs[34,47]. Fifth, it remains to be studied whether our findings generalize to other countries and contexts beyond Germany. While studies on the perceived carbon inequality across income groups[23,24] support our findings for Germany, the distribution of perceived ideal and personal carbon footprints remains unexplored in the literature. Our finding that participants rated their own carbon footprint more favorably likely reflects a self-enhancement tendency, which is typically more pronounced in individualistic cultures (e.g., European Americans) compared to collectivistic ones (e.g., East Asians), as shown by meta-analyses[46,47]. These analyses suggest that self-enhancement motivation is universal but culturally influenced; thus, we expect cultural variations but consistent patterns if our work was replicated in other country settings. The generally ambitious ideal carbon footprint distribution observed in our study resonates with other findings showing broad support for climate action across countries. For instance, Andre et al.[60] conducted a representative survey across 125 countries, interviewing nearly 130,000 individuals, and found that 86% endorsed pro-climate social norms. At the same time, the level of support for climate policies is often underestimated; for instance, 80–90% of survey participants in the United States underestimated the actual support for such policies[61,62] (see also pluralistic ignorance effect[63]).

The present study has three important implications for climate policies and action. First, our findings highlight a broad consensus on the necessity of reducing GHG emissions and ensuring a fair distribution of responsibility. Participants widely agreed that current GHG emissions are far too high and must be significantly reduced, with a common belief that wealthier groups should bear a greater share of the burden rather than all groups contributing equally. These insights challenge the notion that climate action is deeply divisive or that public opinion is polarized. Instead, they reveal a broad consensus on both the urgency of action and the need to ensure fairness. This consensus is particularly important for enhancing support for structural and redistributive climate policies. Raising awareness about the broad agreement on emission reductions and equitable responsibility could strengthen public support for more ambitious policies, countering the misconception that climate action lacks widespread support[60]. This is especially relevant in the EU, including Germany, where policy discussions often emphasize target-setting over ensuring fair contributions. By making fairness a central element of climate policy communication, governments can foster stronger public endorsement and more effective implementation.

Second, our findings demonstrate that individuals have a relatively accurate understanding of actual carbon inequality while estimating their own GHG emissions to be lower compared to others. To address this, policy efforts could target self-perceptions and relative positioning within carbon footprint distributions. However, a key challenge lies in how to correct misperceptions of personal carbon footprints given the robustness of optimistic biases to change[64,65]. Digital tools that provide personalized feedback and educational campaigns tailored to different wealth groups may help to correct biased self-perceptions and foster a greater sense of responsibility by facilitating transparent comparisons. However, as this approach alone may not be sufficient to reduce GHG emissions, a wider policy approach that also focuses on the environment in which individuals decide and act is needed to ensure that sustainable choices are feasible for all and as easy as possible[66].

Third, addressing carbon inequality effectively requires a multi-faceted policy approach that integrates equity, transparency, and targeted support. Emission reduction strategies must be complemented by social equity measures to ensure a fair and socially just transition. For example, redistributing revenues from carbon taxes can be used to promote fairness by supporting less wealthy groups while encouraging wealthier groups to reduce luxury emissions[26]. Policies should also focus on expanding access to sustainable alternatives for less wealthy populations. Additional measures include transparent climate labeling, such as the climate-score label for food products[67], personalized carbon footprint feedback to improve self-perception accuracy, and educational campaigns tailored to different wealth groups. By combining these approaches, policymakers can foster a more equitable and effective response to carbon inequality, ensuring that climate action is both ambitious and socially just.

## Methods

The present investigation is part of the Konstanz Life-Study, a longitudinal multiple-cohort study[68]. The study adhered to the guidelines of the German Psychological Society and the Declaration of Helsinki and was approved by the ethics committee of the University of Konstanz (ID number: 10/2016). Written informed consent was obtained from all participants prior to their participation.

### Participants

Between March 6 and April 29, 2023, we collected survey data from 1415 participants. Of these, 29 participants had to be excluded due to missing values on core variables, leading to a final sample of 1386 participants ($M_{age} = 46.62 \pm 17.97$, range: 18–89; 61.7% women). Participants filled in, onsite and under supervision of trained study staff, a computer-based questionnaire using the software Unipark ($n = 1220$) or a paper-pencil version ($n = 166$).

On average, participants had completed 16.34 ($SD = 2.26$, range: 10–20) years of education. Most participants lived either alone (37.0%) or with one other household member (41.2%), while larger households were less frequent (21.8%). The majority lived with a partner or spouse (57.5%), and about a quarter of households included at least one additional family member (24.7%).

Participants lived in households with a median monthly net income ranging between 3000 and 5000€ (national median monthly net income 2814€ according to the national microcensus 2022[69]) and a median wealth of 33,000€ to 142,999€ (national median net wealth 106,600€ in 2021[70]). The wealth of 40.5% of participants was above the national median (more than 313,000€: 27.2%, and 143,000€ to 313,000€: 13.3%), 22.2% had median assets between 33,000€ and 142,999€, and 35.2% had assets below the national median (2000€ to 32,999€: 27.9%, and below 2000€: 7.3%).

In terms of wealth quintiles, 48.6% ($n = 674$) of the participants perceived themselves as belonging to the middle 20% of the German population. In addition, a greater proportion of participants placed themselves in the upper wealth quintiles of the German population (top 20%: 12.3%, $n = 170$; second 20%: 21.8%, $n = 302$) than in the less affluent wealth quintiles (fourth 20%: 13.9%, $n = 193$; bottom 20%: 3.2%, $n = 45$).

On average, women ($M = 3.13$, $SD = 0.89$) perceived themselves as belonging to a lower wealth quintile than men ($M = 3.47$, $SD = 1.03$, $t(979.8) = -6.21$, $p < 0.001$). Similarly, women ($M = 3.19$, $SD = 1.30$) self-categorized into a lower numeric wealth group than men ($M = 3.36$, $SD = 1.36$, $t(1053.6) = -2.30$, $p = 0.021$). Additionally, household size was positively correlated with both personal wealth quintile ($r = 0.25$, 95% CI [0.20; 0.30], $p < 0.001$) and numeric wealth group ($r = 0.28$, 95% CI [0.23; 0.33], $p < 0.001$). Additional analyses were calculated to

control for gender (Supplementary Tables 2 and 5) and household size (Supplementary Tables 3 and 6).

## Measures and materials

**Importance of a good carbon footprint.** Participants were asked to indicate how important a good carbon footprint is to them on a scale from (1) very unimportant to (5) very important. Specifically, the phrase "gute Klimabilanz (d.h. geringe Treibhausgasemissionen)" was used, translating to "good carbon footprint (i.e., low greenhouse gas emissions)" (see Supplementary Methods for all items). Thus, participants were informed that a good carbon footprint refers to low GHG emissions. In Germany, the terms "Klimabilanz" and "$CO_2$-Fußabdruck" are used interchangeably, both translating to "carbon footprint". The term carbon footprint is widely understood in Germany, and often referenced in the media[71,72].

**Perceptions of ideal and actual carbon footprint distributions in society.** Following Norton and Ariely[13,14] (see also Debbeler et al.[15]), we assessed perceived ideal and actual distributions across five different wealth groups in the German population. Wealth was defined as the sum of all major assets owned in a household (e.g., house, car, cash, savings, shares, etc.) minus any debts (e.g., car loans, mortgages etc.), ensuring that participants referred to the concept of net wealth and approach the task in the same manner. Participants were asked to think about the German population as divided into five quintiles, ranging from the richest to the poorest (i.e., top 20%, second 20%, middle 20%, fourth 20%, bottom 20%).

To assess perceptions of carbon footprint distributions, participants first rated the actual carbon footprint of each of the five wealth quintiles. After evaluating the actual carbon footprint for each wealth quintile in the German population and using the same visual rating scales, participants rated how they believe the carbon footprints of people in the five wealth quintiles should ideally be. Based on previous research[13–15], ideal distributions are defined as the preferred or the desired level of (in-)equality. Comparing ideals with the perceived status quo can provide insight into whether individuals are satisfied with the perceived status quo or desire change[17].

The phrase "Klimabilanz (Treibhausgasemissionen)" was used, translating to "carbon footprint (greenhouse gas emissions)". Thus, while participants were informed that carbon footprint refers to GHG emissions, it was not further defined in terms of GHG emissions attributed to final consumers and production owners. All ratings were made on a visual scale from very bad to very good, respectively. Importantly, a good carbon footprint therefore refers to low levels of GHG emissions, while a bad carbon footprint indicates high GHG emissions—an interpretation we clarify throughout the text. In the computer-based version, the scale ranged from 0 to 100, while in the paper-pencil version, the initial assessment ranged from 0 to 12 cm and was recoded to 0–100. Evaluative ratings (i.e., from very bad to very good) were used rather than numerical judgements (e.g., GHG emissions in $CO_2$-equivalents), as the latter can be more complex and less meaningful for non-experts[32,73–75]. The use of evaluative labels (very bad to very good) rather than quantitative terms (very small to very large) aligns with common language use in Germany, where expressions like "guter" (good) or "schlechter" (bad) carbon footprint are widely recognized in climate discussions. Using familiar evaluative terms directly conveying a specific valence minimizes potential misinterpretation and ensures clarity in respondents' ratings.

For the analyses (except those comparing ratings with objective estimates), ratings were recoded to values between (1) very bad to (5) very good to align with personal carbon footprint ratings (i.e., 1–5) and improve the clarity of descriptive analyses and figures.

For the comparison with objective carbon footprint estimates, participants' carbon footprint ratings (i.e., 0–100) for each of the five wealth quintiles were converted into the relative contribution of each

wealth quintile. Specifically, participants' carbon footprint ratings for each of the five wealth quintiles were reverse-coded, divided by the total sum of these ratings, and converted into percentages.

**Personal carbon footprint.** Participants rated their "own current carbon footprint (greenhouse gas emissions)" on a scale from (1) very bad to (5) very good.

**Objective carbon footprint estimates.** Carbon footprint estimates (i.e., group GHG emission shares) for five wealth groups of the German population (i.e., net wealth quintiles) based on data from 2017. Estimates are based on the consumption approach (i.e., all GHG emissions attributed to final consumers), as well as the mixed and ownership approach (i.e., GHG emissions split between final consumers and production owners) as described in a working paper by Chancel and Rehm[7] and provided by the authors.

**Personal wealth group.** To assess participants' self-categorization of their wealth (i.e., personal wealth quintile), they were asked to place themselves into one of five wealth quintiles in the German population (bottom 20%, fourth 20%, middle 20%, second 20%, top 20%). In addition to identifying their personal wealth quintile, participants were asked to estimate their actual household wealth. For both ratings, the same definition of wealth was provided (see above). They assigned themselves to one of the following five wealth groups (personal numeric wealth group): less than 2000€, 2000 to 32,999€, 33,000 to 142,999€, 143,000 to 313,000€, more than 313,000€. These wealth groups were calculated based on disposable net worth weighted by household size (data from Germany, 2017[76]). The two measures, "personal wealth quintile" and "personal numeric wealth group", were positively correlated ($r = 0.49$, 95% CI [0.45; 0.53], $p < 0.001$, Supplementary Table 9). Since the results for both measures were similar, only the results for "personal wealth quintile" are reported here for brevity, with results for "personal numeric wealth group" provided in the Supplement.

## Statistical analyses
Data was structured using the R packages "data.table" (version 1.16.4)[77], "dplyr" (version 1.1.4)[78], "forcats" (version 1.0.0)[79] and "tidyr" (version 1.3.1)[80]. For data visualizations, the packages "ggplot2" (version 3.5.2)[81], "ggubr" (version 0.6.0)[82], "RColorBrewer" (version 1.1-3)[83], "ggalluvial" (version 0.12.5)[84] and "ggtext" (version 0.1.2)[85] were used. Data were analyzed descriptively with base R and the R package "Rmisc" (version 1.5.1)[86], and with linear regressions (for perceptions of one's own carbon footprint) and multilevel analyses (for perceptions of ideal and actual carbon footprints of five wealth groups). Unstandardized coefficients are reported. Multilevel models were conducted using "lme4"(version 1.1-37)[87] and "lmerTest" (version 3.1-3)[88]. Deviance tests were used to compare models with and without random slopes. Pseudo-$R^2$ values were calculated following Raudenbush and Bryk[89]. All analyses were conducted with R (version 4.4.2)[90].

## Reporting summary
Further information on research design is available in the Nature Portfolio Reporting Summary linked to this article.

## Data availability
The deidentified participant data used in this study are available in the publicly accessible repository KonData under a CC BY 4.0 license at https://doi.org/10.48606/WobQpECnfNuCRarU[91].

## Code availability
The analysis code of this study is available in the publicly accessible repository KonData under a CC BY 4.0 license at https://doi.org/10.48606/WobQpECnfNuCRarU[91].

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

## Acknowledgements

We would like to thank Bettina Ott, Christophe Bousquet, Emma Erhard, Fiona Frank, Magdalena Huber, Verena Klusmann, Fridtjof Nussbeck, Barbara Binder, Anna Adler, Kim Ankermann, Joke Blümke, Veronika Braun, Frauke Drews, Viola Dürbeck, Caroline Eble, Marlies Fuchs, Johanna Gelz, Caroline Götte, Eva Gotzhein, Konstanze Knelles, Ronja Kleß, Lisanne Kley, Isabella König, Marie-Luise Langenberg, Julian Oberwein, Luise Rabe, Ruth Schauenberg, Clio Scholzen, Sophie Schroetter, Paula Späth, Xuehui Sun for their valuable support. This study was funded by the Deutsche Forschungsgemeinschaft (DFG, German Research Foundation) under Germany's Excellence Strategy—EXC 2117–422037984 (the Centre for the Advanced Study of Collective Behaviour; B.R. and H.T.S.), and by the Bundesministerium für Bildung und Forschung (BMBF, Federal Ministry of Education and Research) under 01EL1820A (SMARTACT; B.R. and H.T.S.). The funding sources had no involvement in the study's design; the collection, analysis, and interpretation of data; the writing of the manuscript; and in the decision to submit the manuscript for publication.

## Author contributions

B.R. and H.T.S. developed the study concept. J.E.K., J.K., H.T.S. and B.R. contributed to the study design. J.K., J.E.K. and J.S. conducted the study, including participant recruitment and data collection, under the supervision of B.R. and H.T.S. J.E.K. and J.K. analyzed and interpreted the data with contributions from B.R., H.T.S., J.S., Y.R., L.C. and C.D. L.C. and Y.R. contributed estimates of carbon footprint shares for the five wealth quintiles in the German population. J.K., J.E.K. and B.R. wrote the first draft of the paper. H.T.S., J.S., Y.R., L.C. and C.D. contributed to the writing and revision of the manuscript. J.K. and J.E.K. contributed equally to the manuscript. J.K., J.E.K., J.S., Y.R., L.C., C.D., H.S. and B.R. approved the final version of the manuscript for submission.

## Funding

## Competing interests

The authors declare no competing interests.
