## [Transparent Peer Review file · Nature Communications]

The carbon perception gap in actual and ideal carbon footprints across wealth groups

Corresponding Author: Ms Johanna Köchling

Version 0:

Reviewer comments:

Reviewer #1

(Remarks to the Author)

I congratulate the authors on their revisions. The argument, findings, and contributions are now presented with clarity and precision, greatly enhancing the manuscript.

I especially appreciate the now updated motivation for focusing on Germany, which is well-grounded and contextualized through thoughtful references to comparable studies in other settings. Additionally, the authors have articulated the study's broader relevance admirably well.

I believe this study to be a meaningful contribution to the field.

(Remarks on code availability)

I am not able to run through the whole script.

"Error: object 'actual_cf2' not found" in line 713.

Other than figure 2, I was able to reproduce the findings with the script.

Rather than having a single file with almost 2.000 lines of code, I would have appreciated separated scripts, each performing specific tasks like data wrangling and cleaning, and producing different pieces of the analysis.

Reviewer #2

(Remarks to the Author)

First, I want to thank the authors for carefully considering the comments from the four reviewers, for their clear replies, and for the considerable changes to the manuscript. This is my second review of the manuscript.

A couple of comments concerning my previous review:

-I appreciate the changes to the methods section, the expanded discussion regarding self-serving bias, and the fact that you now explain that the term "carbon footprint" is widely recognized in Germany, as this is likely not the case in all countries.

-I still think that the answer scale (from "very bad" to "very good") for the question regarding how the carbon footprints of people in the five wealth quintiles should ideally be seems weird. To me, it would make more sense with a scale spanning from e.g., low(er) to high(er). However, I understand that you want the same scale for all questions, that the measure is adopted from previous research, and that the discrepancies are the important part.

I do not doubt that the current study could, should, and will be published (also in its current form). However, I do want to repeat some of the comments from the reviewers in round 1, given that Nature Communication is (also) a high-end journal: The study would be stronger if it included "objective" measures of individual carbon footprint alongside the "subjective"

measures, data from several different countries, or an experiment following up on the findings. These comments are still relevant in this review round, but I do understand that it is difficult and not necessarily sensible to add completely new data to the study at this point. I think that, whether the study is sufficient for publishing in this specific journal, in its current form, should be up to the editors to decide.

(Remarks on code availability)

Reviewer #3

(Remarks to the Author)

This study explores perceptions of ideal and actual carbon footprint distributions within society, with a focus on different income groups. It offers insights into public awareness of carbon inequalities and contributes to the understanding of how subjective perceptions align or diverge from objective carbon footprint estimates. However, I have several concerns regarding the study design and its interpretation of findings.

Major concerns:

The major concern lies in the potential bias introduced by the survey design. Respondents might misinterpret the impact of certain behaviors on carbon emissions, leading to skewed results. For example, participants might exaggerate the emissions from using single-use plastic bags, even though such actions typically contribute minimally to their overall carbon footprint. Did the survey design include a detailed list of behaviors and their corresponding carbon emissions to help respondents accurately understand the actual impact of their actions?

Then, the study lacks a clear definition of the term "ideal carbon footprint". While Line 452 references previous literature, the lack of a precise definition caused confusion.

I still feel that this study does not adequately highlight its contributions to the academic field. While the authors addressed an important topic, it does not clearly demonstrate how it fills existing research gaps or advances current knowledge. The policy recommendations presented in the paper remain largely broad and abstract, failing to provide actionable insights. Although the study examines different income groups, it does not offer clear, tailored strategies for emission reduction potential within these groups. For example, incorporating personalized carbon reduction opportunities or identifying high-impact behaviors across income levels could significantly enhance the applicability and relevance of the findings.

Minor comments:

Introduction: The logic of the first paragraph in the introduction is somewhat unclear. For instance, the statement "This is particularly significant as Germany remains the EU's top GHG emitter" lacks sufficient context and does not effectively connect with the broader argument. If the focus is on Germany, providing basic background information about its role in the EU's emissions and climate policies would make the argument more coherent. Alternatively, the content about Germany could be better placed at the end of the introduction, serving as a specific case to emphasize the study's relevance.

Discussion Section: The Discussion section has a lot of room for improvement. While the study addresses an important topic, its logic and focus are not clear enough, resulting in unnecessary repetition and an overall lack of conciseness.

The discussion mentions the comparisons between ideal, actual, and objective carbon footprints multiple times but does not clearly articulate how these concepts are related. A unified analysis of their comparisons and implications, rather than repeating fragmented insights, would enhance the section's clarity.

The analysis of key topics, such as individuals' optimistic perceptions of their carbon footprints, is scattered across the section and lacks depth. The authors might consider explaining more about the main mechanisms behind these optimistic perceptions and the implications for public attitudes toward climate policies (e.g., reduced support for policies targeting individual emissions).

The policy recommendations remain overly broad and abstract, making it difficult to connect them to the study's findings or practical applications. You might need to address specific strategies, such as implementing differentiated carbon taxes for higher-income groups or designing carbon feedback tools to increase public awareness of their own emissions, or clearly link the recommendations to the research's theoretical contributions and practical implications, explaining how these policies could be implemented and what challenges they might address.

(Remarks on code availability)

I have checked the availability of uploaded code and data.

Version 1:

Reviewer comments:

Reviewer #2

(Remarks to the Author)

I see that one of my comments in the last round of review was incorrect and misleading, and you might want to change/remove the sentence you added to the discussion because of it.

I wrote: "I still think that the answer scale (from "very bad" to "very good") for the question regarding how the carbon footprints of people in the five wealth quintiles should ideally be seems weird. To me, it would make more sense with a scale spanning from e.g., low(er) to high(er)".

You have now added a comment regarding this to the discussion: "The use of evaluative labels ("very bad" to "very good") rather than quantitative terms ("very low" to "very high") aligns with common language use in Germany, where expressions like "guter" (good) or "schlechter" (bad) carbon footprint are widely recognized in climate discussions. Using familiar evaluative terms directly conveying a specific valence minimizes potential misinterpretation and ensures clarity in respondents' ratings."

I recognize now that the scale would span from e.g., "smaller" to "bigger/larger", not from "lower" to "higher" as I wrote in my comment, since the question is about carbon footprints and not emissions.

Thus, I don't think the sentence you added to the discussion makes sense and I would either change or delete it.

* My comment pointed back to a comment I made in the first round of reviews:

"I am unsure about the question regarding ideal carbon footprints, and what it tells us. You write that "participants were asked to rate how the carbon footprints (GHG emissions) of people in the five wealth quintiles should ideally be. Ratings were assessed as evaluative (i.e., from "very bad" to "very good)". Why would not everyone ideally want all groups to have "very good" carbon footprints? What would be the motivation to wish for a bad or moderately bad ideal carbon footprint? ..In sum, I am not sure what this question measures or contributes, except for illustrating that most respondents care about mitigating climate change. To make it more relevant, I wish I knew what the respondents believe a "very good" carbon footprint would entail for the different wealth quintiles."

My main point in the last round, which I failed to convey, was that it would make more sense to answer the question about ideal carbon footprints in relative terms.

If the question is something like: "How should the carbon footprint for people in this group ideally be?"

People could, for example, answer on a scale from:

Much smaller than it is today -> Smaller than it is today -> The same as today -> Larger than it is today -> Much larger than it is today.

E.g.: "The carbon footprint for people in this group should ideally be smaller than it is today", "The carbon footprint for people in this group should ideally be larger than it is today".

My issue with the current answer scale is that I find it weird to answer the question: "How should the carbon footprint for people in this group ideally be?" with "very bad".

But again, I understand that you cannot change this now, and that you wanted the same scale for all your questions.

I have no further comments, and I do not need to review the paper again.

(Remarks on code availability)

Reviewer #3

(Remarks to the Author)

I think the authors have adequately addressed my previous concerns, and the storyline now flows very well. I have no further comments.

(Remarks on code availability)

I have reviewed the code. The authors have provided an R file along with a README file containing clear instructions for installation and running the application. Additionally, the data is available in CSV format. Overall, the code is well-organized and the results of the paper appear to be reproducible.

Reviewer #1 (Remarks to the Author):

I congratulate the authors on their revisions. The argument, findings, and contributions are now presented with clarity and precision, greatly enhancing the manuscript.

I especially appreciate the now updated motivation for focusing on Germany, which is well-grounded and contextualized through thoughtful references to comparable studies in other settings. Additionally, the authors have articulated the study's broader relevance admirably well.

I believe this study to be a meaningful contribution to the field.

We thank the reviewer for the encouraging feedback and valuable comments, which have contributed to this improved version of the manuscript. We are pleased to hear that the revision has enhanced the clarity and precision of the manuscript.

Re #1: I am not able to run through the whole script.

"Error: object 'actual_cf2' not found" in line 713.

Other than figure 2, I was able to reproduce the findings with the script.

Thank you for bringing this to our attention. This error occurred during the restructuring of the code and does not affect any of the results. We have now corrected the code.

Re #2: Rather than having a single file with almost 2.000 lines of code, I would have appreciated separated scripts, each performing specific tasks like data wrangling and cleaning, and producing different pieces of the analysis.

Thank you for this suggestion. We have now submitted separate scripts to enhance the clarity of the code.

Reviewer #2 (Remarks to the Author):

First, I want to thank the authors for carefully considering the comments from the four reviewers, for their clear replies, and for the considerable changes to the manuscript. This is my second review of the manuscript.

A couple of comments concerning my previous review:

-I appreciate the changes to the methods section, the expanded discussion regarding self-serving bias, and the fact that you now explain that the term "carbon footprint" is widely recognized in Germany, as this is likely not the case in all countries.

-I still think that the answer scale (from "very bad" to "very good") for the question regarding how the carbon footprints of people in the five wealth quintiles should ideally be seems weird. To me, it would make more sense with a scale spanning from e.g., low(er) to high(er). However, I understand that you want the same scale for all questions, that the measure is adopted from previous research, and that the discrepancies are the important part.

I do not doubt that the current study could, should, and will be published (also in its current form). However, I do want to repeat some of the comments from the reviewers in round 1, given that Nature Communication is (also) a high-end journal: The study would be stronger if it included "objective" measures of individual carbon footprint alongside the "subjective" measures, data from several different countries, or an experiment following up on the findings. These comments are still relevant in this review round, but I do understand that it is difficult and not necessarily sensible to add completely new data to the study at this point. I think that, whether the study is sufficient for publishing in this specific journal, in its current form, should be up to the editors to decide.

Thank you very much for your encouraging feedback on our revised manuscript. We greatly appreciate your positive comments and the time you have taken to review our work once again.

We are pleased to hear that the adjustments made to the methods section and the expanded discussion on self-serving bias have been well-received. Regarding the answer scale ("very bad" to "very good") for the question on how the carbon footprints of people in the five wealth quintiles should ideally be, we acknowledge your continued reservations. We appreciate your suggestion to use a scale ranging from low(er) to high(er) instead of good/bad. We understand the rationale behind your recommendation and would like to elaborate on our approach and the considerations that led to our choice:

(1) As you correctly pointed out, using the same scale across the triad of actual, ideal, and self reduces measurement errors compared to employing different scales. Consistency in measurement is crucial to ensure comparability and minimize bias in the assessment of the carbon perception gap.

(2) The choice of scale relates to the distinction between direct and indirect methods of assessing the ideal footprint distribution. The direct method involves respondents directly comparing the ideal and actual carbon footprints on a shared rating scale (e.g., higher/lower or better/worse). This method is commonly used in social psychology and bias research, as it provides a straightforward comparison between two components, such as the ideal and actual footprint distributions. However, we have chosen to employ the indirect method, wherein respondents evaluate the actual and ideal footprint distributions using two separate but identical scales. This approach allows for the independent assessment of each

component before deriving comparisons between them. Indirect methods have also been extensively used in research and have demonstrated reliability and validity (see review by Shepperd, Klein, Waters, & Weinstein, 2013; Zell et al., 2019). While both methods have their merits, our decision to use the indirect approach was influenced by several considerations. The direct method is indeed more efficient, requiring fewer items and reducing participant workload. However, it may also introduce greater social desirability bias, as participants might adjust their responses to align with perceived study goals or present themselves more favorably (Shepperd et al., 2013; Zell et al., 2019). This potential bias can affect the accuracy of the data. In contrast, the indirect method, though slightly more complex and time-consuming, offers the advantage of generating a richer dataset. By assessing the actual and ideal footprint distributions separately, it becomes possible to calculate the perception gap more precisely and explore variations in the way respondents perceive both constructs.

(3) Regarding the wording of the answer scale, we opted for the labels ("very bad" to "very good") instead of "very low" to "very high" for two reasons. As detailed in our previous response, the term carbon footprint is widely recognized in Germany and commonly used in media coverage of climate change and public discussion about climate action. Evaluative terms such as "gute" (good) or "schlechte" (bad) "Klimabilanz" or "CO₂-Fußabdruck," both translating to "carbon footprint", are standard in this context. Therefore, using the terms bad/good aligns with typical language use in Germany.

In addition, we would like to argue that the use of such evaluative measures has the advantage of directly conveying a specific valence (very good/very bad), and thus it may have been more directly clear to participants whether the label referred to a climate-friendly or climate-unfriendly option.

We have added this line of reasoning to the methods section (lines 502-507) and included a paragraph in the limitations section (lines 367-374). Additionally, we have adapted the figure legends by providing additional descriptors (i.e. low and high GHG emissions) to improve the clarity of our labels (i.e., "very good" and "very bad") for readers. Furthermore, in the discussion section we acknowledge the value of objective measures of individual carbon footprints for future research (lines 337-341).

Reviewer #3 (Remarks to the Author):

This study explores perceptions of ideal and actual carbon footprint distributions within society, with a focus on different income groups. It offers insights into public awareness of carbon inequalities and contributes to the understanding of how subjective perceptions align or diverge from objective carbon footprint estimates. However, I have several concerns regarding the study design and its interpretation of findings.

Major concerns:

Re #1 The major concern lies in the potential bias introduced by the survey design. Respondents might misinterpret the impact of certain behaviors on carbon emissions, leading to skewed results. For example, participants might exaggerate the emissions from using single-use plastic bags, even though such actions typically contribute minimally to their overall carbon footprint. Did the survey design include a detailed list of behaviors and their corresponding carbon emissions to help respondents accurately understand the actual impact of their actions?

We thank the reviewer for highlighting concerns regarding potential bias arising from the survey design and misperceptions of behaviors and associated carbon emissions, which provides an opportunity to further clarify our methodology and findings.

(1) While we did not supply a comprehensive list of behaviors with their corresponding GHG emissions or additional information that could influence respondents' perceptions, we argue that, in real-world decision-making, individuals often operate under conditions of incomplete information and limited knowledge. Therefore, even if potentially imprecise or biased, these perceptions play a significant role in shaping behavioral choices and policy support.

(2) Furthermore, we would like to clarify that potential biases stemming from survey design and misperceptions of behaviors and GHG emissions represent two distinct sources of bias: structural bias from survey design and cognitive or motivational bias from respondents. While structural bias cannot be entirely excluded, it seems unlikely because we used an established questionnaire method (e.g., Debbeler et al., 2021; Norton & Ariely, 2011, 2013), and the response patterns did not suggest misinterpretation or varying interpretations related to question wording. Given the strong consistency in the pattern of results, it is highly likely that the findings reliably and accurately reflect respondents' perceptions.

(3) In addition, participants' perceptions did not appear generally biased. The perceived distribution of actual carbon footprints closely aligned with objective estimates derived from the consumption-based approach. This suggests that participants had a fairly accurate understanding of how emissions are distributed among wealth groups in Germany, despite the absence of additional information on the carbon impact of specific behaviors. This alignment further supports the validity of the observed response pattern.

(4) If participants had consistently exaggerated or misperceived emissions, we would expect weaker alignment with objective data and less consensus in ratings of actual and ideal carbon distribution. Instead, our findings suggest a differential bias, particularly evident in positively skewed perceptions of personal carbon footprints.

(5) The consistent tendency for participants to rate their personal carbon footprint more favorably—regardless of wealth—points to a self-serving motivational bias, likely involving selective focus on climate-friendly behaviors when assessing one's personal carbon footprint. For example, wealthier groups may emphasize recycling, while lower-income groups may focus on reduced travel (Wynes et al., 2020). Such selective information activation and retrieval is well-documented in psychological research (e.g., Ecker, 2022). Therefore, our results suggest that respondents did not generally misperceive carbon footprints due to the survey design but rather exhibited a positively skewed perception of

their own footprint, potentially driven by self-enhancement motives and selective attention to specific climate-friendly behaviors.

(6) Studies show that such self-related optimistic biases are common and that they can be resistant to correction, even when objective information is provided (e.g., Aue et al., 2021; Chowdhury et al., 2014; Kuzmanovic et al., 2015; Sharot et al., 2011). Correcting such misperceptions, for instance through (repeated) personalized feedback, remains an area of research with mixed results (e.g., Dreijerink & Paradies, 2020). However, we are not aware of studies specifically examining the effect of personalized feedback on societal carbon footprint perceptions. As suggested in the previous review round, it seems plausible that personalized feedback could lead individuals to lower their perceived ideal contributions when realizing they fall short of their expectations.

We have expanded on this reasoning in the revised Discussion and included a paragraph in the limitations section (lines 355-367).

References

- Aue, T., Dricu, M., Moser, D. A., Mayer, B., & Bührer, S. (2021). Comparing personal and social optimism biases: Magnitude, overlap, modifiability, and links with social identification and expertise. *Humanities and Social Sciences Communications*, 8(1), 1–12. <https://doi.org/10.1057/s41599-021-00913-8>
- Chowdhury, R., Sharot, T., Wolfe, T., Düzel, E., & Dolan, R. J. (2014). Optimistic update bias increases in older age. *Psychological Medicine*, 44(9), 2003–2012. <https://doi.org/10.1017/S0033291713002602>
- Kuzmanovic, B., Jefferson, A., & Vogeley, K. (2015). Self-specific optimism bias in belief updating is associated with high trait optimism. *Journal of Behavioral Decision Making*, 28(3), 281–293. <https://doi.org/10.1002/bdm.1849>
- Sharot, T., Korn, C. W., & Dolan, R. J. (2011). How unrealistic optimism is maintained in the face of reality. *Nature Neuroscience*, 14(11), 1475–1479. <https://doi.org/10.1038/nn.2949>

Re #2 Then, the study lacks a clear definition of the term "ideal carbon footprint". While Line 452 references previous literature, the lack of a precise definition caused confusion.

Regarding the definition of the term "ideal carbon footprint," we acknowledge the need for clarity. Based on previous research (e.g., Debbeler et al., 2021; Norton & Ariely, 2011, 2013), "ideal distributions" are defined as the preferred or the desired level of (in-)equality. We asked respondents to indicate the carbon footprint they thought would be ideal for each of the quintiles, starting with the top 20% and ending with the bottom 20%.

We have added this clarification to the method section (Lines 489 - 492).

Re #3 I still feel that this study does not adequately highlight its contributions to the academic field. While the authors addressed an important topic, it does not clearly demonstrate how it fills existing research gaps or advances current knowledge. The policy recommendations presented in the paper remain largely broad and abstract, failing to provide actionable insights. Although the study examines different income groups, it does not offer clear, tailored strategies for emission reduction potential within these groups. For example, incorporating personalized carbon reduction opportunities or identifying high-impact behaviors across income levels could significantly enhance the applicability and relevance of the findings.

We thank the reviewer for the valuable feedback. Our core finding—that people consistently believe emissions are far too high, should be significantly reduced, yet perceive themselves as contributing more than others—forms the three pillars of the carbon perception gap. Notably, participants' perceptions of actual carbon inequality closely aligned with estimates from the consumption-based approach, indicating an accurate understanding of wealth-based disparities in GHG emissions. To our knowledge, this framing is novel and the revisions now clarify our contributions and offer actionable insights for policymakers (Discussion, lines 391-429).

Minor comments:

Re # 4 Introduction: The logic of the first paragraph in the introduction is somewhat unclear. For instance, the statement "This is particularly significant as Germany remains the EU's top GHG emitter" lacks sufficient context and does not effectively connect with the broader argument. If the focus is on Germany, providing basic background information about its role in the EU's emissions and climate policies would make the argument more coherent. Alternatively, the content about Germany could be better placed at the end of the introduction, serving as a specific case to emphasize the study's relevance.

Thank you very much for your feedback regarding the clarity and structure of the first paragraph in the introduction. We have carefully considered your suggestions and, in line with feedback from Reviewer 1, have made adjustments to improve the coherence of our argument.

While we have chosen to retain Germany in the first paragraph rather than moving it to the end of the introduction, we have clarified its relevance more explicitly. Germany was highlighted in this context due to its combination of high wealth inequality, status as the EU's top GHG emitter, and ambitious climate goals to reduce emissions by at least 65% by 2030. This combination creates substantial challenges for equitable climate action, particularly given Germany's influential role as a key economic and political player in the EU. To strengthen this point, we added clarifying sentences (Introduction, lines 36-39).

Re #5 Discussion Section: The Discussion section has a lot of room for improvement. While the study addresses an important topic, its logic and focus are not clear enough, resulting in unnecessary repetition and an overall lack of conciseness.

The discussion mentions the comparisons between ideal, actual, and objective carbon footprints multiple times but does not clearly articulate how these concepts are related. A unified analysis of their comparisons and implications, rather than repeating fragmented insights, would enhance the section's clarity.

The analysis of key topics, such as individuals' optimistic perceptions of their carbon footprints, is scattered across the section and lacks depth. The authors might consider explaining more about the main mechanisms behind these optimistic perceptions and the implications for public attitudes toward climate policies (e.g., reduced support for policies targeting individual emissions).

The policy recommendations remain overly broad and abstract, making it difficult to connect them to the study's findings or practical applications. You might need to address specific strategies, such as implementing differentiated carbon taxes for higher-income groups or designing carbon feedback tools to increase public awareness of their own emissions, or clearly link the recommendations to the research's theoretical contributions and practical implications, explaining how these policies could be implemented and what challenges they might address.

Thank you for giving us the opportunity to improve the Discussion. We have now substantially restructured and revised the Discussion with a focus on clarity and conciseness, while further elaborating on the policy implications (please also see #Re 3).

Reviewer #3 (Remarks on code availability):

I have checked the availability of uploaded code and data.

Point-by-point response

Reviewer #2 (Remarks to the Author):

I see that one of my comments in the last round of review was incorrect and misleading, and you might want to change/remove the sentence you added to the discussion because of it.

I wrote: “I still think that the answer scale (from “very bad” to “very good”) for the question regarding how the carbon footprints of people in the five wealth quintiles should ideally be seems weird. To me, it would make more sense with a scale spanning from e.g., low(er) to high(er)”.

You have now added a comment regarding this to the discussion: “The use of evaluative labels (“very bad” to “very good”) rather than quantitative terms (“very low” to “very high”) aligns with common language use in Germany, where expressions like “guter” (good) or “schlechter” (bad) carbon footprint are widely recognized in climate discussions. Using familiar evaluative terms directly conveying a specific valence minimizes potential misinterpretation and ensures clarity in respondents’ ratings.”

I recognize now that the scale would span from e.g., “smaller” to “bigger/larger”, not from “lower” to “higher” as I wrote in my comment, since the question is about carbon footprints and not emissions.

Thus, I don’t think the sentence you added to the discussion makes sense and I would either change or delete it.

** My comment pointed back to a comment I made in the first round of reviews:*

“I am unsure about the question regarding ideal carbon footprints, and what it tells us. You write that “participants were asked to rate how the carbon footprints (GHG emissions) of people in the five wealth quintiles should ideally be. Ratings were assessed as evaluative (i.e., from “very bad” to “very good”)”. Why would not everyone ideally want all groups to have “very good” carbon footprints? What would be the motivation to wish for a bad or moderately bad ideal carbon footprint? ..In sum, I am not sure what this question measures or contributes, except for illustrating that most respondents care about mitigating climate change. To make it more relevant, I wish I knew what the respondents believe a “very good” carbon footprint would entail for the different wealth quintiles.”

My main point in the last round, which I failed to convey, was that it would make more sense to answer the question about ideal carbon footprints in relative terms.

If the question is something like: “How should the carbon footprint for people in this group ideally be?”

People could, for example, answer on a scale from:

Much smaller than it is today -> Smaller than it is today -> The same as today -> Larger than it is today -> Much larger than it is today.

E.g.: “The carbon footprint for people in this group should ideally be smaller than it is today”, “The carbon footprint for people in this group should ideally be larger than it is today”.

My issue with the current answer scale is that I find it weird to answer the question: “How should the carbon footprint for people in this group ideally be?” with “very bad”.

But again, I understand that you cannot change this now, and that you wanted the same scale for all your questions.

I have no further comments, and I do not need to review the paper again.

We sincerely thank the reviewer for their thoughtful follow-up and for taking the time to clarify their earlier comment.

We greatly appreciate your insights and your helpful reflection on the previous suggestion. In line with your current recommendation, we have revised the sentence accordingly. To ensure consistency and clarity, we carefully reviewed the manuscript and revised the labeling. To avoid any possible confusion, we would also like to clarify that the sentence you referred to was included in the *Methods* section, rather than the *Discussion* section.

Our decision to assess both actual and ideal distributions of carbon footprints was informed by prior research on preferences for distributions in other domains, notably work by Norton and colleagues on wealth distribution, as well as research by Debbeler et al. (2021) on health disparities. These studies show that while people generally endorse equality, their ideals often incorporate certain forms of inequality that are perceived as fair or even desirable, especially when they reflect contextual constraints or moral justifications.

Applied to the context of carbon emissions, this suggests that individuals may not necessarily endorse uniformly “very good” carbon footprints (i.e., low emissions) across all wealth groups, but instead endorse differentiated ideals based on contextual considerations. For instance, higher emissions for lower-income groups may be perceived as more acceptable due to their limited financial and temporal resources, which constrain their ability to reduce emissions. In contrast, higher standards may be applied to wealthier groups, who are seen as more capable of making reductions. Thus, an ideal distribution that includes some variation—even including “moderately bad” carbon footprints for certain groups—can still reflect an overall preference for fairness and sustainability. This nuanced reasoning is consistent with our empirical findings and helps explain why the ideal is not necessarily one of uniform minimal emissions across all segments of society.

Furthermore, regarding our methodological choice, both a direct approach (e.g., asking if footprints should be smaller or larger) and an indirect approach (as used here) offer specific advantages. We employed an indirect method, using the same scale to assess both actual and ideal carbon footprints, which enables a more precise calculation of perception gaps by independently measuring each construct. Importantly, consistent with the aforementioned literature, ideal distributions elicited through such measures are commonly interpreted as a contrast to the status quo—even without an explicit temporal reference such as “today.” This interpretation is particularly likely in our case given the question order: participants evaluated actual carbon footprints immediately before reporting their ideals. This sequencing likely reinforced the contrastive nature of the ideal judgments, supporting our interpretation that they reflect a desire for change.

Once again, we are very grateful for your constructive feedback and your thoughtful engagement throughout the review process.

Reviewer #3 (Remarks to the Author):

I think the authors have adequately addressed my previous concerns, and the storyline now flows very well. I have no further comments.

Reviewer #3 (Remarks on code availability):

I have reviewed the code. The authors have provided an R file along with a README file containing clear instructions for installation and running the application. Additionally, the data is available in CSV format. Overall, the code is well-organized and the results of the paper appear to be reproducible.

We thank the reviewer for the positive evaluation of the adapted storyline and would like to thank them for their very helpful comments.